# Biswas–Chatterjee–Sen Model on Solomon Networks with Two Three-Dimensional Lattices

**DOI:** 10.3390/e26070587

**Published:** 2024-07-10

**Authors:** Gessineide Sousa Oliveira, Tayroni Alencar Alves, Gladstone Alencar Alves, Francisco Welington Lima, Joao Antonio Plascak

**Affiliations:** 1Dietrich Stauffer Computational Physics Laboratory, Departamento de Física, Universidade Federal do Piauí, Teresina 64049-550, PI, Brazil; sousagessi21@gmail.com (G.S.O.); tay@ufpi.edu.br (T.A.A.); 2Departamento de Física, Universidade Estadual do Piauí, Teresina 64002-150, PI, Brazil; alves.gladstone@gmail.com; 3Departamento de Física, Centro de Ciências Exatas e da Natureza (CCN), Universidade Federal da Paraíba, Cidade Universitária, João Pessoa 58051-970, PB, Brazil; pla@uga.edu; 4Departamento de Física, Universidade Federal de Minas Gerais, C. P. 702, Belo Horizonte 30123-970, MG, Brazil; 5Department of Physics and Astronomy, University of Georgia, Athens, GA 30602, USA

**Keywords:** non-equilibrium, phase transition, Monte Carlo simulations, Solomon networks

## Abstract

The Biswas–Chatterjee–Sen (BChS) model of opinion dynamics has been studied on three-dimensional Solomon networks by means of extensive Monte Carlo simulations. Finite-size scaling relations for different lattice sizes have been used in order to obtain the relevant quantities of the system in the thermodynamic limit. From the simulation data it is clear that the BChS model undergoes a second-order phase transition. At the transition point, the critical exponents describing the behavior of the order parameter, the corresponding order parameter susceptibility, and the correlation length, have been evaluated. From the values obtained for these critical exponents one can confidently conclude that the BChS model in three dimensions is in a different universality class to the respective model defined on one- and two-dimensional Solomon networks, as well as in a different universality class as the usual Ising model on the same networks.

## 1. Introduction

In the last four decades, there has been a great increase in the study of the general behavior of people in a community. From a theoretical point of view, this rather complex dynamical system must, certainly, take into account that people can not only influence their neighbors but be influenced by them as well. According to this scenario, some early models have been introduced by Stauffer [1] and Galam [2,3] using basically local majority rule arguments, which are also now called dynamics of opinions. Other models, just to cite some examples, are the Ising model (IM) [4,5], the majority-vote model (MVM) [6], and the Biswas, Chatterjee, and Sen (BChS) [7,8] model. Regarding the Ising model, it is worth mentioning that, despite being originally proposed to treat magnetic systems, almost nine decades before the new ideas by Stauffer and Galam, the IM is so versatile that it can also be employed in the study of other interdisciplinary fields as well (see, for instance, Refs. [9,10,11,12]).

In general, the dynamical properties of the models mentioned above are better studied by considering an underlined regular lattice or network, in which every site of the lattice or node of the network are occupied by an individual (sometimes also called an agent). In this way, within each lattice or network, one has the proper type of mutual interactions that are defined by the corresponding model under study, e.g., Ising model, MVM, BChS model, and others.

In the particular case of the BChS model, which was introduced in 2012 and is the subject of the present study, the pair interactions are allowed to be either positive or negative. The interaction signs are modeled by a single noise parameter *q*, that represents the fraction of negative interactions. In addition, the opinion variables can be made either continuous or discrete. This non-equilibrium system has been studied from Monte Carlo simulations on different regular lattices as well as different networks. Regardless of being continuous or discrete, the model presents a second-order phase transition at $q_{c}$, with the noise parameter *q* playing the role of temperature in ordinary magnetic phase transitions. For $q<q_{c}$, the system has a non-zero order parameter, while for $q>q_{c}$, the order parameter is zero. However, contrary to what happens in ordinary magnetic transitions, the exponents and the universality class of the model depend on the particular chosen lattice or network. The reader is directed to Ref. [13] for a detailed relation of the universality class of the BChS model on different lattices and networks. Ref. [14] also presents a recent review of the progress already made in the study of the BChS model.

A rather more realistic situation has been proposed by Sorin Solomon [15,16], by considering two different lattices, where one lattice reflects one kind of environment (for instance, home) and the other lattice a different environment (for instance, workplace). In general, one labels the sites in the workplace lattice *i* and a random permutation P(i) of the order already established in this lattice will provide the sites in the home lattice. In such real situations, the neighbors at home differ from the neighbors in the workplace (except, of course, when everybody works at home). Such constructions are called Solomon networks (SNs) [9,17,18], reflecting the fact that each individual is equally shared by two lattices, just as in the King Solomon biblical story. As a result, the net interaction of the relevant variables defined at site *i* is a sum of the corresponding interactions of site *i* with its neighboring sites on the workplace lattice, plus the interactions with the neighbor sites of P(i) on the home lattice. Some correlation between home and workplace lattices can be introduced into the model by choosing P(i) not completely at random. For example, people selecting a workplace closer to their homes. However, generally only random permutations have been considered in P(i) in order not to proliferate additional theoretical parameters. The increase in the connectivity of each site *i* makes the SNs close to small-world networks [19,20].

The majority of the studies involving SNs have employed two linear chains and two square lattices, implying one-dimensional (1D) and two-dimensional (2D) topologies, respectively. However, three-dimensional (3D) lattices can be important as well, and not only from a theoretical point of view. For instance, one could make the correspondence of environments in the country as 1D, in towns as 2D, and in big cities, due to their building verticalization in some areas, as 3D. This certainly implies that people living in such different environments present, as is indeed the case, different behaviors regarding the way they live and work.

The BChS model has already been studied in different lattices and networks [13] and also in 1D and 2D SNs [21]. Concerning the SNs, the critical exponents of the second-order phase transition indicate that the universality class of the model is dependent on the dimensionality of the lattices, as is expected from general renormalization group arguments. It should be stressed here that from the results one has from the literature it is not possible to definitely assert that a particular model on a particular network will undergo a first- or second-order transition, or even any transition at all. It is thus worthwhile treating the BChS model in 3D SNs, by defining two three-dimensional lattices, namely, simple cubic ones for simplicity. It will be seen that the model does indeed belong to a different universality class, in agreement with renormalization ideas. Moreover, the hyperscaling relation is satisfied for dimensions smaller than three, meaning that the upper critical dimensionality is certainly greater than or equal to four [22].

Thus, in the present work, the BChS dynamical system on SNs has been studied through Monte Carlo simulations, allied with finite-size scaling techniques, considering three-dimensional simple cubic lattices. The plan of the paper is the following. In the next section, the BChS model in discrete opinion dynamics is defined. In this same section, the corresponding Monte Carlo simulation details and the respective *thermodynamic* quantities used to obtain the critical behavior are presented. The results are discussed in Section 3 and some concluding remarks are summarized in the last section.

## 2. Model and Simulations

In this section, the BChS model is briefly defined, in its discrete form, and the thermodynamic-like variables, from which one can obtain the phase transition properties, are presented. Some details of the employed Monte Carlo simulations, together with the finite-size scaling relations of the relevant quantities to treat the model on finite-size lattices of linear size *L* are also discussed.

### 2.1. Biswas–Chatterjee–Sen Model and Relevant Thermodynamic-like Variables

The BChS model has already been described and previously studied in different regular lattices and networks (see, for instance, Refs. [7,8,21,23]). Below, we have just the main ingredients of the model directed specifically to SNs. We will keep here the same notation, for the relevant variables, already used in the previous studies of this model in other contexts.

Agents are set on each *i*-th node of an SN with *N* sites. At time step *t*, the agents have opinion variables defined by $o_{i}\left(t\right)$, that can assume three different values: −1, 0, or +1. The BChS rules adopted here for updating $o_{i}\left(t\right)$ are as follows.

(i)Begin by constructing an initial configuration, where for each site *i* of the SN one of the three opinion states is randomly assigned. This is the configuration of the opinion variables $o_{i}$ at time t=0, and can be labeled as $o_{i}\left(t=0\right)=o_{i}$.(ii)The workplace lattice is sequentially swept for every site *i* to be updated. At this point one has $o_{i}\left(t\right)=o_{i}$, where all $o_{i}$ are the corresponding opinion variables that have been updated at time t−1.(iii)For a given site *i*, one of its nearest neighbors is randomly selected and an affinity $\mu_{ij}$ is ascribed for this bond. This affinity parameter is another discrete variable that assumes a value +1, but can be turned negative with a probability *q*. This probability *q* acts as an external noise, modeling local discordances.(iv)The opinion variables of both sites *i* and *j* are then assigned new values as
(1)$$\begin{array}{ccc} o_{i}+\mu_{ij}o_{j} & \to & o_{i}\;, \end{array}$$
(2)$$\begin{array}{ccc} o_{j}+\mu_{ij}o_{i} & \to & o_{j}\;. \end{array}$$
where on the right-hand sides $o_{i}$ and $o_{j}$ are the updated opinion states.(v)Another site *ℓ*, randomly distant $|\vec{R}|$ from site *i*, is additionally selected and an affinity $\mu_{i\ell }$ is ascribed for this bond in the same way as before.(vi)The opinion variable of these sites are now updated as
(3)$$\begin{array}{ccc} o_{i}+\mu_{i\ell }o_{\ell } & \to & o_{i}\;, \end{array}$$
(4)$$\begin{array}{ccc} o_{\ell }+\mu_{i\ell }o_{i} & \to & o_{\ell }\;. \end{array}$$(vii)If the opinion state turns out to be outside the interval [−1,+1], it is automatically made $o_{i}=+1$ if $o_{i}>+1$, or $o_{i}=-1$ if $o_{i}<-1$.(viii)One sweep through the lattice constitutes one Monte Carlo step (MCS) from time *t* to t+1. Thus, one has $o_{i}\left(t+1\right)=o_{i}$, with $o_{i}$ the updated set of opinion variables from items (iii)–(vii).

Note that using only the workplace lattice and making a two-step selection comprising a nearest neighbor *j* and a distant site *ℓ* is a simplified, and less computer-time consuming, version of the original two-lattice SN [9].

A convenient order parameter *O* for this model consists of averaging the opinion variables $o_{i}\left(t\right)$ over all individuals as
(5)$$O=\left|\sum_{i=1}^{N}o_{i}\left(t\right)\right|/N,$$
where *t* is chosen to be large enough for the system having reached the stationary state and *N* is the total number of sites of the SN and given by $N=L^{3}$. In ordinary magnetic systems one has a thermal driven phase transition. In the present model, one has a kind of random-configuration-driven phase transition instead. The phase transition means that for $q<q_{c}$, O≠0 and an ordered phase is established, while for $q>q_{c}$, O=0 and the system is in a disordered phase. Exactly at $q=q_{c}$, a second-order phase transition takes place.

In order to fully categorize this phase transition, in addition to the above order parameter, it is also convenient to analyze its magnetic-like variables; for instance, the order parameter fluctuation $O_{f}\left(q\right)$ or susceptibility (which is the analogous to the magnetic susceptibility) and its reduced fourth-order Binder cumulant $O_{4}\left(q\right)$. These observables are formally expressed as
(6)$$\begin{array}{ccc} O(q) & = & [{\langle O\rangle }_{t}{]}_{av}, \end{array}$$
(7)$$\begin{array}{ccc} O_{f}\left(q\right) & = & N[{\langle O^{2}\rangle }_{t}-{\langle O\rangle }_{t}^{2}{]}_{av}, \end{array}$$
(8)$$\begin{array}{ccc} O_{4}\left(q\right) & = & 1-[\frac{{\langle O^{4}\rangle }_{t}}{3{\langle O^{2}\rangle }_{t}^{2}}{]}_{av}, \end{array}$$
where ${\langle \cdots \rangle }_{t}$ stands for different time averages (which are computed after the system has reached the stationary state), and ${\left[\cdots \right]}_{av}$ means the averages taken over different initial configurations. The simulation details of what was used for obtaining the above quantities are given below.

### 2.2. Monte Carlo Simulations and Finite-Size Scaling Relations

The Monte Carlo simulations were performed following the algorithm described in items (i)–(viii) above on SNs of finite sizes *L*. The sizes of the lattices were chosen as L=6, 8, 10, 12, 14, 16, 18, up to 20, with $N=L^{3}$ total sites. The relevant quantities in Equations (6)–(8) were computed as a function of the noise probability, also called the noise parameter, *q*. Close to the transition region, the noise parameter steps were chosen as Δq=0.001.

In all simulations, to obtain the relevant observables we performed several MCSs on the SNs. We simulated 120 SN realizations for each network size to make quench averages. For each network replica, we considered $10^{5}$ MCSs to let the system evolve to a stationary state, and then, another $10^{5}$ MCSs to collect $10^{5}$ values of the opinion variables to measure the desired observables. Error bars were calculated by using the jackknife resampling technique [24,25].

Since all the variables defined in Equations (6)–(8) depend, in addition to the noise parameter *q*, also on the SN sizes, the criticality exhibited by the model can be inferred by analyzing their behavior as a function of *L* and close to the transition point $q_{c}$. Thus, according to the finite-size scaling hypothesis one has, for large system sizes *L*, the following power-law expressions of the relevant variables (for further details one can see, for instance, Refs. [24,26]):(9)$$\begin{array}{ccc} O(q) & = & L^{-\beta /\nu }f_{O}\left(x\right), \end{array}$$(10)$$\begin{array}{ccc} O_{f}\left(q\right) & = & L^{\gamma /\nu }f_{O_{f}}\left(x\right), \end{array}$$(11)$$\begin{array}{ccc} O_{4}\left(q\right) & = & f_{O_{4}}\left(x\right), \end{array}$$(12)$$\begin{array}{ccc} q_{c}\left(L\right) & = & q_{c}+cL^{-1/\nu }, \end{array}$$
where β, ν, and γ are the critical exponents of the order parameter, correlation length, and fluctuation of the order parameter, respectively; $f_{k}\left(x\right)$, with $x=L^{1/\nu }\left(q-q_{c}\right)$ and $k=\{O,\;O_{f},\;O_{4}\}$, are the respective scaling functions. In the above equations it is assumed that *q* is close to the corresponding critical noise $q_{c}$ for an infinite-size network. In addition, for each finite network of *L* size, we can also estimate its pseudo-critical noise parameter $q_{c}\left(L\right)$ by looking, for example, at the peak presented by the fluctuation in the order parameter as a function of *q* or cumulant crossings for different network sizes. In this case, one can use Equation (12), where *c* is a non-universal constant.

It is implicit in Equation (11) that the order parameter cumulants, at the critical point, have no dependence on the size *L*. As a result, all cumulants should cross at the same value of the noise parameter $q_{c}$. However, when the finite lattices are not large enough, residual corrections to finite-size scaling make the crossing points of the cumulants suffer a systematic shift as *L* varies. In this case, there is a useful, although not so frequently employed, way of extrapolating the critical point from the cumulant crossings through an analysis of the scaling of $q_{c}\left(L\right)$ with the lattice sizes *L* and $L_{s}$. From phenomenological renormalization group ideas, Nightingale [27] has shown that an extrapolation procedure, where the scaling factor is given by $ln(L/L_{s})$, furnishes quite good results when correlation length on strips of different widths is used. This same strategy has also proved to be suitable when considering the cumulant crossings for different lattice sizes [28,29,30,31]. The scaling form can be expressed as
(13)$$q_{c}\left(L\right)=q_{c}+c^{\prime }/\ln\left(L/L_{s}\right),$$
where $q_{c}\left(L\right)$ is the value of the noise parameter at the cumulant crossing of the lattice *L* and some reference lattice size $L_{s}$, with $L_{s}<L$, and $c^{\prime }$ is another non-universal constant. The above equation is convenient when one does not know the correlation length critical exponent that is necessary in Equation (12).

## 3. Results and Discussion

The general behavior of the relevant quantities for this model, namely, *O*, $O_{f}$, and $O_{4}$, given by Equations (Equation 6)–(8), are, respectively, displayed in the top panels, (a), (b), and (c), of Figure 1. The different lattice sizes used in the present work are listed in the legend. In these panels, only the lines of the data are shown for a better visualization of the behavior one has as a function of the noise parameter *q*. The details of extracting the critical properties of the model from these data are discussed below.

The fourth-order Binder cumulant of the order parameter $O_{4}$, as a function of the disorder parameter *q* close to the phase transition, is depicted in Figure 2 for several SNs having different numbers of nodes. One can clearly see from this figure that the system undergoes a second-order phase transition, since the cumulants tend to cross at the same value, corresponding to the critical disorder parameter $q_{c}$ [26]. In the axes scales used in Figure 2, one can make a rough estimate of the critical noise and the universal value of the Binder cumulant, $q_{c}=0.195\left(3\right)$ and $O_{4}^{*}=0.301\left(5\right)$, respectively. In the above data, the computed errors are statistical ones and for questions of clarity are not shown in Figure 2.

However, looking closely at the interception region of the cumulants in Figure 2, it is noticed that the crossings do not occur at the same place due to still finite-size effects. This region has been magnified in the inset of Figure 3. Considering as a reference lattice $L_{s}=6$, given by the thicker line in the inset, one can see that due to finite-size effects the value of the noise parameter at the crossings increases as the lattice size *L* systematically increases. Using these crossings as estimates of $q_{c}\left(L\right)$, and plotting them as a function of $1/\ln(L/L_{s})$, one obtains the data shown in the main graph of Figure 3. The errors on $q_{c}\left(L\right)$ have been, in this case, estimated by using the crossings of $O_{4L}-\Delta O_{4L}$ with $O_{4L_{s}}+\Delta O_{4L_{s}}$, where $\Delta O_{4L}$ and $\Delta O_{4L_{s}}$ are the respective errors on $O_{4L}$ and $\Delta O_{4L_{s}}$ obtained via the jackknife procedure. It is not surprising that the data are not all aligned, because quite small lattices have been considered. However, these data look similar to previous ones obtained for the Ising and Heisenberg models [28,29,30,31]. Using Equation (Equation 13) for the larger lattices L≥14, when one expects that the large lattice scaling regime has been reached, one obtains $q_{c}=0.1950\left(7\right)$ and $O_{4}^{*}=0.304\left(2\right)$ in the thermodynamic limit. These are more precise values for the transition disorder parameter and for the cumulant of the model than the rough estimate obtained from just inspecting the crossings in Figure 2. Larger values of $L_{s}$ could also be considered, but the data turn out to be more disperse, making it more difficult to obtain a reasonable fit. The critical noise for the present 3D SNs is listed in Table 1 together with the results for 1D and 2D networks for comparison.

Having in hand a good estimate of the critical noise probability, Equation (Equation 9) can be used to evaluate the critical exponent ratio β/ν by computing the order parameter at $q_{c}$ for different lattice sizes. Figure 4 depicts the ln–ln plot of the average opinion at the critical disorder, $O(q_{c})$, as a function of the system sizes *L* considered herein. In this case, it is easy to see that, from Equation (Equation 9), the magnitude of the slope of the linear fit gives the critical exponent ratio β/ν. The corresponding result β/ν=0.789(3) is also reproduced in Table 1 and is clearly different from the values obtained for the same model on SNs in one and two dimensions.

The very same procedure can be easily extended to the order parameter fluctuation using Equation (10). Figure 5 displays the ln–ln plot of the order parameter fluctuation (susceptibility) at the critical disorder value, $O_{f}\left(q_{c}\right)$, at the estimated $q_{c}$, as a function of the lattice size *L*. From Equation (10) one obtains the ratio γ/ν as the slope of the linear fit to the data. The fit provides γ/ν=1.312(8) and is listed in Table 1 for a comparison to the values for other dimensions.

It is worthwhile to note that Equation (10) can still be used to obtain the value of the ratio γ/ν by considering the maximum value of the order parameter fluctuation, $O_{f}\left(q_{max}\right)$ that occurs at $q=q_{max}$. It turns out the $q_{max}$ is close, but not the same, as $q_{c}$; see Figure 1b that shows the behavior of $O_{f}$ close to the transition region. Figure 5 also displays the ln–ln plot of $O_{f}\left(q_{max}\right)$ as a function of the lattice size *L*. In this case, one obtains γ/ν=1.372(50). The error in this exponent ratio is larger, but still comparable to that obtained from the estimate at $O(q_{c})$. Moreover, both exponent ratios are different from the previous estimates on 1D and 2D networks.

As mentioned in the last paragraph, the value of $q_{max}$, where the fluctuation in the order parameter displays a maximum, is different from $q_{c}$. For the present model, $q_{max}$ approaches the infinite limit $q_{c}$ from below, as is apparent in Figure 1b. In fact, this noise probability is generally interpreted as a new $q_{c}\left(L\right)$ from which the transition point $q_{c}$ is obtained by extrapolating Equation (12) to the thermodynamic limit. Of course, in this case, one needs to know the correlation length exponent ν. However, if one adopts here a different path instead, and uses $q_{c}\left(L\right)$ for all values of *L* in Equation (12), with the already estimated $q_{c}$, one is able to compute the exponent 1/ν as the magnitude of the slope of the linear fit to the data. Such a procedure is shown in Figure 6, that displays the ln–ln plot of $|q_{c}\left(L\right)-q_{c}|$ as a function of *L*. The respective correlation length critical exponent is given by 1/ν=1.63(10) and also displayed in Table 1.

Finally, in Figure 1, the data collapse of the rescaled order parameter $OL^{\beta /\nu }$ is shown in panel (d), the rescaled fluctuation of the order parameter $O_{f}L^{-\gamma /\nu }$ in panel (e), and the rescaled reduced Binder cumulant $O_{4}$ in panel (f), all as a function of the rescaled probability displacement $\left(q-q_{c}\right)L^{1/\nu }$. As in the top panels, just the lines of the actual Monte Carlo data have been used for a better evaluation of the collapse and the accuracy of the critical quantities. The corresponding lattice sizes *L* are listed in the legend of (d), where only larger sizes L≥10 have been considered in the data collapse. It can be seen that the data collapse presents an excellent agreement with scaling relations given in Equations (Equation 9), (10) and (12), mainly for $q>q_{c}$ and even close to $q≲q_{c}$, which indicates that the evaluation of the critical exponent ratios β/ν, γ/ν, and 1/ν are reasonably accurate. Moreover, as can be noticed from the last column of Table 1, they obey, within the error bars, the hyperscaling relation for all lattice dimensionalities.

## 4. Concluding Remarks

The Biswas–Chatterjee–Sen model, in its discrete version and defined on three-dimensional Solomon networks, has been studied through extensive Monte Carlo simulations for several values of the local consensus controlling parameter. From the data collapse displayed in panels (d)–(f) of Figure 1, and from the scaling behavior depicted in Figure 4, Figure 5 and Figure 6, it is clear that the model really undergoes a well-defined second-order phase transition.

It is clear that the present SN sizes do not ensure that the large-size scaling regime of Equations (Equation 9), (10), and (12) has been achieved. In fact, we have seen that finite-size effects are actually present, for instance, in the fourth-order Binder cumulants in Figure 3. This means that corrections to scaling could be implemented in the scaling relation. This will certainly require extra unknown correction-to-scaling exponents. However, the evaluation of the critical exponents seems to be reasonable, even for the SN sizes employed in Figure 4, Figure 5 and Figure 6, since the data are quite aligned for a proper linear fit (the same in Figure 3, but only for the largest lattices).

In addition, looking at Figure 1d,e for the *O* and $O_{f}$ variables, one can clearly see the finite-size behavior of the data collapse. While for $q>q_{c}$ all sizes collapse to the same curve, for $q<q_{c}$ the collapse systematically tends to the same curve as the SN size increases. For the two largest SNs, namely, L=18 and L=20, all data are almost superimposed in the whole range of $\left(q-q_{c}\right)L^{1/\nu }$. This means that we have reached the large-size regime for the universal scaling functions of the variables defined in Equations (Equation 6)–(8). It is also evident from Figure 1f that the reduced fourth-order Binder cumulant $O_{4}\left(q\right)$ turns out to be a more robust variable, because there has been no finite-size effects for all SN sizes used herein.

From the critical exponents of the BChS model on SNs, conveyed in Table 1, it becomes apparent that the model in three dimensions is indeed in a different universality class to the model defined in one and two dimensions. This is an expected result and in agreement with the general trend coming from renormalization group arguments. Here, we also emphasize that our results are different from the three-dimensional BChS model on regular lattices in Ref. [8]. Despite the critical exponent ν being, within the error bars, compatible to the Ising universality class at 3D, this is not true for the other exponents, which could be affected by strong finite-size effects. The present results thus indicate that the BChS model has its own universality class.

It is also interesting to note that although the critical exponents have a sensitive change with the dimension of the network, the critical noise probabilities, responsible for the phase transition in the model, seem not to vary much as the connectivity of the networks is increased by the dimensionality of the lattices.

Finally, an additional study of the BChS model on four-dimensional SN networks should be very interesting from the pure theoretical point of view. Recall that the Ising model has a lower critical dimension 1D and an upper critical dimension 4D. The BChS model on SNs has a lower critical dimension 0D and the present results clearly indicate, if such arguments are valid here, an upper critical dimension greater than 3D. The question of violation of the hyperscaling relation and a possible unified view of the finite-size scaling valid in all dimensions should also be addressed in this case [22].

## Figures and Tables

**Figure 1 entropy-26-00587-f001:**
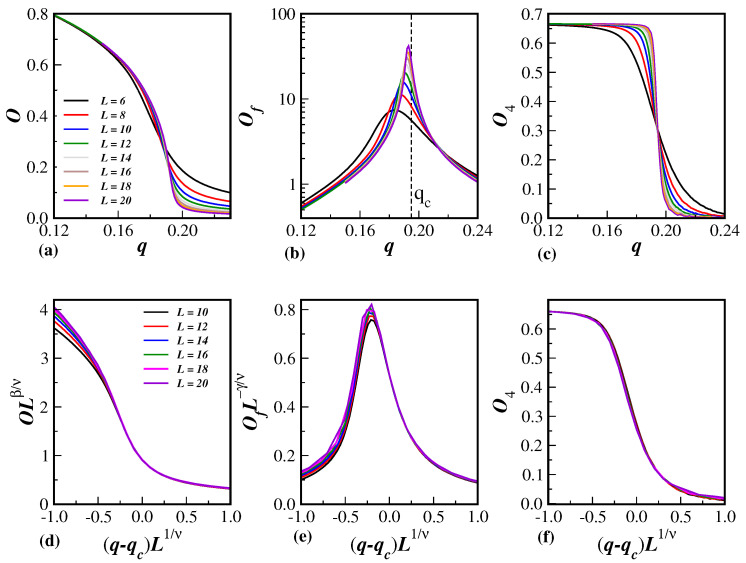
The panels (**a**–**c**) show the order parameter *O*, the fluctuation of the order parameter $O_{f}$, and the Binder fourth-order cumulant $O_{4}$ as a function of the noise parameter *q*. The dashed line in (**b**) locates the estimated critical noise $q_{c}$. The legends in (**a**) also apply to (**b**,**c**). The panels (**d**–**f**) show the corresponding data collapse of the quantities displayed in the top panels using the finite-size scaling relations given in Equations (Equation 9), (10), and (12) combined with numerical data in Table 1. The legends in (**d**) also apply to (**e**,**f**).

**Figure 2 entropy-26-00587-f002:**
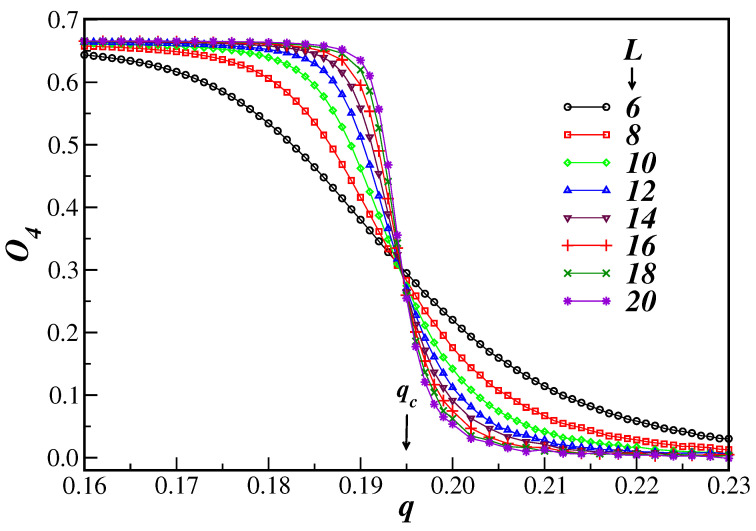
Fourth-order Binder cumulant of the order parameter $O_{4}$, plotted as a function of the disorder parameter *q*, for several values of the lattice sizes *L*. The vertical arrow indicates the corresponding estimate of $q_{c}=0.195\left(3\right)$. For clarity, only the general trend in the cumulant behavior is shown and the error bars have been omitted.

**Figure 3 entropy-26-00587-f003:**
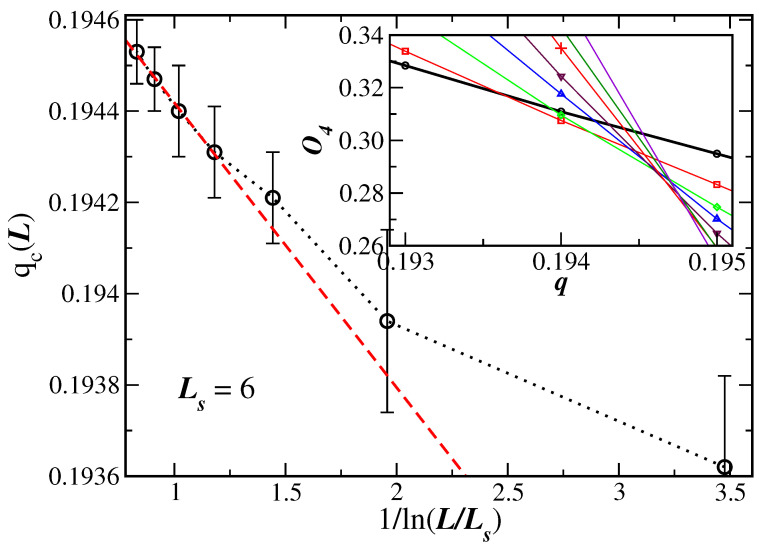
Cumulant disorder parameter crossings $q_{c}\left(L\right)$, as a function of $1/\ln(L/L_{s})$, for several values of *L*, considering the reference lattice $L_{s}=6$ (main graph). The dotted line is just a guide to the eyes and the dashed line a linear fit according to Equation (Equation 13) considering the larger lattices L≥14. The inset gives a magnified view close to the interception region of Figure 2. The legend of Figure 2 also applies to this inset and a thicker line (joining the open circles) for the reference lattice $L_{s}=6$ is used.

**Figure 4 entropy-26-00587-f004:**
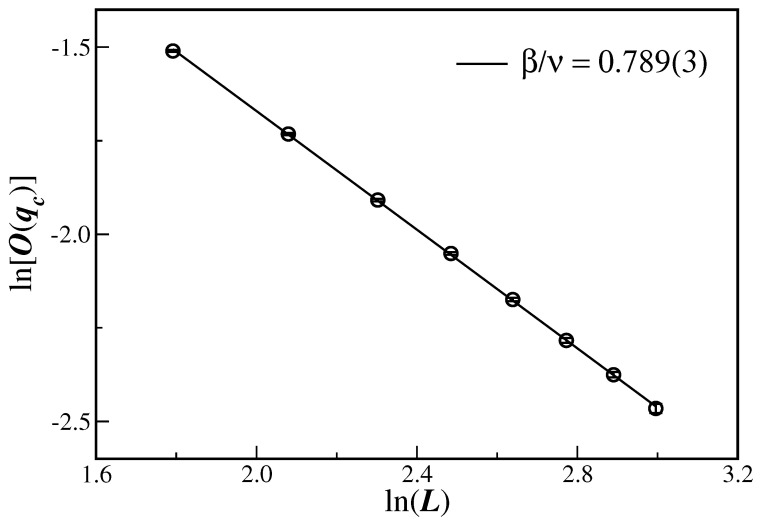
Ln–ln plot of the average opinion at the estimated critical disorder $O(q_{c})$ as a function of the logarithm of the system sizes *L*. The line is the best linear fit according to Equation (Equation 9), with the slope being the critical exponent ratio β/ν given in the legend. Note that the shown error bars are smaller than the symbol sizes.

**Figure 5 entropy-26-00587-f005:**
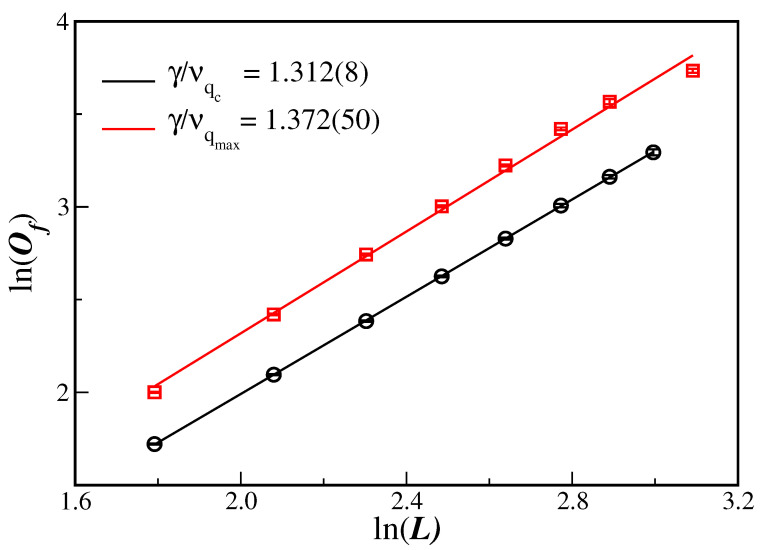
Ln–ln plot of the susceptibility $O_{f}\left(q_{c}\right)$ at the estimated $q_{c}$ and maximum value $O_{f}\left(q_{max}\right)$ as a function of the system sizes *L*. The lines are the best linear fit with the slope being the critical exponent ratio γ/ν. The displayed error bars are smaller than the symbol sizes.

**Figure 6 entropy-26-00587-f006:**
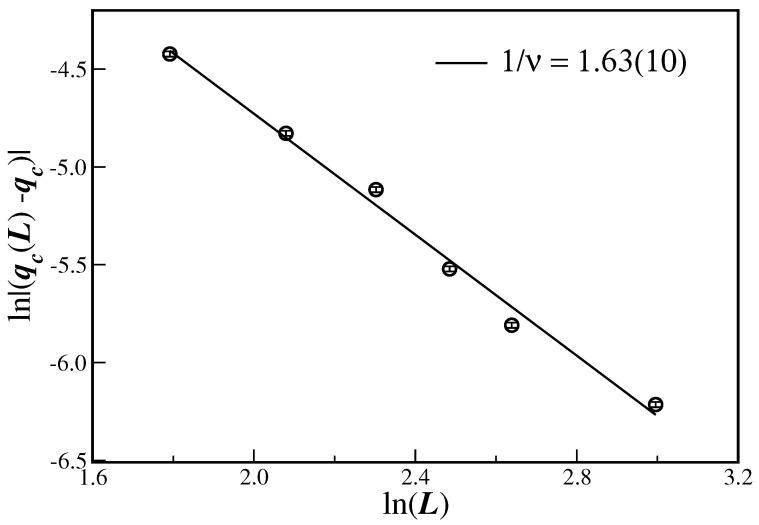
Ln–ln plot of the magnitude of the displacement $|q_{c}\left(L\right)-q_{c}|$, with $q_{c}\left(L\right)$ extracted from Figure 1b. The line is a linear fit with the corresponding exponent given in the legend.

**Table 1 entropy-26-00587-t001:** Critical noise parameter $q_{c}$ and critical exponent ratios 1/ν, β/ν, γ/ν of the BChS model on three-dimensional SNs (3D). Also shown, in the two first rows, are the previous results obtained for the 1D and 2D on the same networks from Ref. [21] for a comparison. The exponent ratio $\gamma /\nu (q_{max})$ is the result from the maximum of the order parameter susceptibility. The last column gives the hyperscaling relation. Error bars are statistical only.

	$q_{c}$	1/ν	β/ν	$\gamma /\nu (q_{c})$	$\gamma /\nu (q_{\max})$	d=2β/ν−γ/ν
1D	0.215(2)	0.52(5)	0.238(7)	0.511(5)	0.524(6)	1.00(2)
2D	0.216(2)	0.92(4)	0.53(4)	1.02(4)	1.06(7)	2.1(1)
3D	0.1950(7)	1.63(10)	0.789(3)	1.312(8)	1.372(50)	2.95(6)

## Data Availability

The data presented in this study are available on request from the corresponding author.

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
