# Peer review of "Biswas–Chatterjee–Sen Model on Solomon Networks with Two Three-Dimensional Lattices"

_entropy, 2024, doi:10.3390/e26070587_

Round 1

Reviewer 1 Report

Comments and Suggestions for Authors

The manuscript deals with the opinion dynamics within the Biswas-Chatterjee-Sen model. It is discussed on Solomon network, in a version based on two three dimensional lattices. The applied methodology and the results are similar to the previous works published by some of the Authors, but -- to my best knowledge -- the results are original since the considered underlying lattice (three dimensional, simple cubic) has never been considered for the Biswas-Chatterjee-Sen model so far. Therefore I believe that the manuscript is suitable for publication after some corrections and clarifications which I suggest below.

1. Line 19: The Ising model is mentioned in such a way it looks like a suggestion that it was proposed as one of the models for opinion dynamics. I suppose that this in not what the Authors want to say.

2. Description of the used algorithm given in the lines 95-118 in Sec. 2.1 needs to be improved, since in my opinion it is very difficult to understand what the Authors mean at certain points. First of all, it is written that the rules are given for 'updating (...) to t+1', so every time when the time advances, but I suppose that the first step '(i) Begin by...' which requires random assignments of opinion states is performed only once, at t=0, and not repeated every time when the simulation goes from t to t+1. Then, the step (iii): does that mean that 'j' is selected randomly from all neighbors of 'i', those from the workplace lattice and from the home lattice, with equal probabilities of choosing one of those lattices first, or rather with equal probabilities of choosing any of the neighbours, taking all of them as one set from which the choice is made? In the final note of the step (v), it is noted that 'one sometimes updates sites that are not necessarily neighbors and can even be far apart': again it may suggest that 'i' and 'j' are not neighbors on both lattices, but it seems to contradict (iii), at least if 'sharing a bond' means the same thing as being neighbors. Finally, in the step (vi) I would also suggest to be very precise and clearly state that o(t+1)>+1 becomes +1, and o(t+1)<-1 becomes -1, if this is what Authors do at such situations.

3. Line 126: What does it mean that ordered and disordered phases 'become equal'?

4. Line 133: I would also like to suggest explaining the statement that equations (4), (5) and (6) have been 'somehow prepared' to be used in Monte Carlo simulations.

5. In lines 177-178 it is written that panels (a)-(c) of Fig.1 show O, O_f and O_4 'given by Eqs. (7)-(9)'. Shouldn't it rather be 'given by Eqs. (4)-(6)', which define those quantities?

6. Lines 194 and Figure 3: the term 'thicker line' is used and I guess the Authors refer here to the black line with circles for the points. It is not clear, since the line is maybe only marginally thicker than other lines, if at all. I would rather suggest referring to the color and type of symbols.

7. Figure 3: In the inset, what is the meaning of the colors? (Probably the same L's as in Figure 2?)

8. Figure 3: How the error values shown here with the error bars were estimated for q_c(L)?

9. Table 1: In the header and caption of Table 1 'q_c^max' is used, while in the main text there is only 'q_max'. Are they the same thing?

10. References are written in somewhat random style: sometimes with titles of papers, sometimes without. Additionally, for [26] 'submitted': is at available e.g. as a preprint? If so, I would suggest to add a reference to the preprint server.

Comments on the Quality of English Language

English is fine most of the time, with some small mistakes or typos only (itens-->items, stablished-->established, underlined-->underlying).

Author Response

“Summary of the changes made and response to the referees’
reports on entropy-3061821”
Mr. Uros Simic
Assistant Editor, MDPI
Dear Editor,
Thank you for your email sending us the reports from the referees. We would also like
to deeply thank the referees for their reports and for the constructive criticism on our
manuscript.
In order to address the comments by the referees, we have made some clarifications in
the body of the manuscript. We hope we have appropriately answered all the Referees’
questions.
We summarize below the changes we have made in the text, as well as a brief response
to all concerns or criticisms of the Referees. They are all written in red, both in the revised
manuscript and the text below.
Sincerely,
F. W. Lima (on behalf of all authors)

I.
REPORT REVISOR 1
The manuscript deals with the opinion dynamics within the Biswas-Chatterjee-Sen model.
It is discussed on Solomon network, in a version based on two three dimensional lattices.
The applied methodology and the results are similar to the previous works published by
some of the Authors, but – to my best knowledge – the results are original since the consid-
ered underlying lattice (three dimensional, simple cubic) has never been considered for the
Biswas-Chatterjee-Sen model so far. Therefore I believe that the manuscript is suitable for
publication after some corrections and clarifications which I suggest below.
We thank the reviewer for a positive evaluation of our paper.

11. Line 19: The Ising model is mentioned in such a way it looks like a suggestion that it
was proposed as one of the models for opinion dynamics. I suppose that this in not what
the Authors want to say.
The reviewer is right, and we have made a comment, in the end of the first paragraph
of the Introduction, that this model was originally proposed for studying magnetic phase
transitions. Refs. 10-12 have also been included.

2. Description of the used algorithm given in the lines 95-118 in Sec. 2.1 needs to be
improved, since in my opinion it is very difficult to understand what the Authors mean at
certain points. First of all, it is written that the rules are given for ’updating (...) to t+1’,
so every time when the time advances, but I suppose that the first step ’(i) Begin by...’
which requires random assignments of opinion states is performed only once, at t=0, and
not repeated every time when the simulation goes from t to t+1. Then, the step (iii): does
that mean that ’j’ is selected randomly from all neighbors of ’i’, those from the workplace
lattice and from the home lattice, with equal probabilities of choosing one of those lattices
first, or rather with equal probabilities of choosing any of the neighbors, taking all of them as
one set from which the choice is made? In the final note of the step (v), it is noted that ’one
sometimes updates sites that are not necessarily neighbors and can even be far apart’: again
it may suggest that ’i’ and ’j’ are not neighbors on both lattices, but it seems to contradict
(iii), at least if ’sharing a bond’ means the same thing as being neighbors. Finally, in the
step (vi) I would also suggest to be very precise and clearly state that o(t + 1) > +1 becomes
+1, and o(t + 1) < −1 becomes -1, if this is what Authors do at such situations.
After re-reading the algorithm, we have agreed that the text led indeed to the confusions
raised by the referee. In fact, the description of algorithm was incomplete. We have thus
rewritten the algorithm on page 3, addressing all of the above comments. We hope this new
description of the algorithm is now clearer for the reader.

3. Line 126: What does it mean that ordered and disordered phases ’become equal’ ?
In order to avoid confusion and extra possible discussions out of the scope of the present
subject, we have dropped this comment from the paper, as is now in line 133 of the new
manuscript. Nevertheless, if the referee is interested on this kind of viewing a second-order
or continuous phase transition, please go to Ref. Entropy 24 (1) (2022), 63 that has more
details, including multicritical points.

4. Line 133: I would also like to suggest explaining the statement that equations (4), (5)
2and (6) have been ’somehow prepared’ to be used in Monte Carlo simulations.
These equations are now numbered (6), (7) and (8). We have extended the discussion on
how they are used to get the corresponding quantities, just after these new equations, on
lines 140-143 of the new manuscript.

5. In lines 177-178 it is written that panels (a)-(c) of Fig.1 show O, O f and O 4 ’given by
Eqs. (7)- (9)’. Shouldn’t it rather be ’given by Eqs. (4)-(6)’, which define those quantities?
We thank the referee for pointing us out this mistake. It has been corrected with the new
numbers (6)-(8).

6. Lines 194 and Figure 3: the term ’thicker line’ is used and I guess the Authors refer
here to the black line with circles for the points. It is not clear, since the line is maybe only
marginally thicker than other lines, if at all. I would rather suggest referring to the color
and type of symbols.
We agree that the line was not thick enough to be distinguished from the others. We
have thicken this line in the new Fig. 3 and also referred it to the open circles in the legend.

7. Figure 3: In the inset, what is the meaning of the colors? (Probably the same L’s as
in Figure 2?)
Yes, the same as Fig. 2. It is now stated in the legend.

8. Figure 3: How the error values shown here with the error bars were estimated for
q c (L)?
The errors in the previous version have been indeed underrated. We have done new
estimates that are in the new Figure 3. The explanation on how the errors have been
obtained is highlighted in red in lines 208-210 of the new manuscript.

9. Table 1: In the header and caption of Table 1 q c max is used, while in the main text
there is only q max . Are they the same thing?
Thanks for pointing us out this mistake because they are not the same. It has been
corrected in the header and caption of Table 1 as q max .

10. References are written in somewhat random style: sometimes with titles of papers,
sometimes without. Additionally, for [26] ’submitted’: is at available e.g. as a preprint? If
so, I would suggest to add a reference to the preprint server.
We have now fully referenced the (new) Ref. [28]. We have also uniformized the references
according the journal style (not red highlighted in the text).

3II.
ASSISTANT EDITOR’S COMMENTS
1. Academic/Guest Editor comments: To make the paper fit the scope of our SI well,
please add some content to link the work to the scope of the issue and to Ralph’s work
(mentioning Ralph’s papers in the list of references, or maybe speculating on what will
happen to the model in higher dimensions), to ensure its consideration in this SI.
In the Introduction (lines 79-81) and in the last section (lines 308-310), we have added
some discussion about the violation or not of the hyperscaling relation in higher dimensions
and also included a new reference 22 of the Ralph’s work.

2. Technical comments: During the technical check of your manuscript, we noticed
that a high proportion of the cited references belong to you or your co-authors: Refs.
8,13,14,17,18,19,20, which is a self-citation rate of about 27 Could you please check whether
the inclusion of each of these references is appropriate? As MDPI is a member of COPE
(https://publicationethics.org), all references in our published articles must contribute to
the scholarly content of the paper and avoid bias (self-citations, journal citations, school of
thought, etc.,) and reflect the current state of knowledge in the field. We encourage you to
consider this and reduce the self-citations to make sure that only the most relevant citations
are kept.

We have reduced two self-citations and included additional references that are all red
highlighted in the revised manuscript.
4

Reviewer 2 Report

Comments and Suggestions for Authors

In the present manuscript the authors use Monte Carlo simulations and finite-size scaling analysis to identify and characterise the second-order transition in the  Biswas-Chatterjee-Sen model of opinion dynamics in on three-dimensional Solomon networks. These models of opinion dynamics are interesting in the field of statistical physics stretched to social dynamics and networks and therefore the current work has the merit to be published in Entropy. 

I list below some comments for the authors to consider prior to publication:

 (i) I am not sure that terms “critical social temperature” and “social phase transition” are meaningful.

(ii) In the numerical scheme more information is needed, especially for equilibration of the systems with increasing size, as well the definition of statistical accuracy, i.e., how errors have been computed and why they appear so small; naively this looks like strong correlations of the numerical data.

(iii) Eq. (11) needs a bit more explanation to my eyes and reasoning as to where it comes from.

(iv) In all finite-size scaling relations the corrections term is missing; I suspect the lattice sizes are not big enough to allow for such a determination but I think a parallel analysis should be included or at least critically discussed.

(v) In all fittings showed (including the data collapse) there is no relevant discussion about fitting quality.

(vi) The authors claim as a main result that the critical exponents of this continuous transition change with the dimensionality; I don’t think this is an unexpected result but on the other hand it is definitely expected. So I am not sure if this can be taken as the main message from the paper; I would expect on the contrary some critical discussion about possible matching of the universality class of this system with well-known models of statistical physics. For example, the critical exponent \nu that the authors quote is within error bars compatible to the Ising universality class at 3D (but this is not true for the other exponents which could be affected by strong finite-size effects).

(vii) What about the universal value of the Binder cumulant at the crossing point?

Author Response

“Summary of the changes made and response to the referees’
reports on entropy-3061821”
Mr. Uros Simic
Assistant Editor, MDPI
Dear Editor,
Thank you for your email sending us the reports from the referees. We would also like
to deeply thank the referees for their reports and for the constructive criticism on our
manuscript.
In order to address the comments by the referees, we have made some clarifications in
the body of the manuscript. We hope we have appropriately answered all the Referees’
questions.
We summarize below the changes we have made in the text, as well as a brief response
to all concerns or criticisms of the Referees. They are all written in red, both in the revised
manuscript and the text below.
Sincerely,
F. W. Lima (on behalf of all authors)

I.
REPORT REVISOR 2
Comments and Suggestions for Authors
In the present manuscript the authors use Monte Carlo simulations and finite-size scaling
analysis to identify and characterise the second-order transition in the Biswas-Chatterjee-
Sen model of opinion dynamics in on three-dimensional Solomon networks. These models
of opinion dynamics are interesting in the field of statistical physics stretched to social
dynamics and networks and therefore the current work has the merit to be published in
Entropy. I list below some comments for the authors to consider prior to publication:
(i) I am not sure that terms “critical social temperature” and “social phase transition”
are meaningful.
1We have dropped these terms from the abstract and rephrased accordingly.

(ii) In the numerical scheme more information is needed, especially for equilibration of
the systems with increasing size, as well the definition of statistical accuracy, i.e., how errors
have been computed and why they appear so small; naively this looks like strong correlations
of the numerical data.
The second paragraph on section 2.2 of the revised manuscript, lines 151-156, gives more
information about how the quantities have been computed and the numerical errors evalu-
ated.

(iii) Eq. (11) needs a bit more explanation to my eyes and reasoning as to where it comes
from.
We have included some brief comments on how now Eq. (13) has been proposed. The
comments are just before Eq. (13) on lines 177-182 of the revised manuscript

(iv) In all finite-size scaling relations the corrections term is missing; I suspect the lattice
sizes are not big enough to allow for such a determination but I think a parallel analysis
should be included or at least critically discussed.
That is true. The lattice sizes are not large enough to assure the scaling relations without
corrections to finite size. We could see that in Fig.3. However, Figs. 4-6 seem to furnish
reasonable linear fits considering all lattices. The new Fig1 shows, in addition, the conver-
gence of the universal scaling function for lattices L = 18 and 20. It seems thus that for the
two largest lattices the scaling regime has been reached. We have added a new second and
third paragraph in the final section with the above comments (lines 273-288).

(v) In all fittings showed (including the data collapse) there is no relevant discussion
about fitting quality.
We believe that the discussion that is now conveyed in the second and third paragraph
of the revised version address to this important point as well.

(vi) The authors claim as a main result that the critical exponents of this continuous
transition change with the dimensionality; I don’t think this is an unexpected result but
on the other hand it is definitely expected. So I am not sure if this can be taken as the
main message from the paper; I would expect on the contrary some critical discussion
about possible matching of the universality class of this system with well-known models of
statistical physics. For example, the critical exponent ν that the authors quote is within
error bars compatible to the Ising universality class at 3D (but this is not true for the other
2exponents which could be affected by strong finite-size effects).
We included such discussion in the new fourth paragraph of the final section (lines 291-
297). Our results indicate that the BChS model has its own universality class, with the
hyperscaling relation being valid, within the error bars, for all dimensions (see, please, new
Table 1).

(vii) What about the universal value of the Binder cumulant at the crossing point?
Thanks for pointing us out this detail that was missing in our old version. The corre-
sponding values are given now in lines 202 and 207.

II.
ASSISTANT EDITOR’S COMMENTS
1. Academic/Guest Editor comments: To make the paper fit the scope of our SI well,
please add some content to link the work to the scope of the issue and to Ralph’s work
(mentioning Ralph’s papers in the list of references, or maybe speculating on what will
happen to the model in higher dimensions), to ensure its consideration in this SI.
In the Introduction (lines 79-81) and in the last section (lines 308-310), we have added
some discussion about the violation or not of the hyperscaling relation in higher dimensions
and also included a new reference 22 of the Ralph’s work.

2. Technical comments: During the technical check of your manuscript, we noticed
that a high proportion of the cited references belong to you or your co-authors: Refs.
8,13,14,17,18,19,20, which is a self-citation rate of about 27 Could you please check whether
the inclusion of each of these references is appropriate? As MDPI is a member of COPE
(https://publicationethics.org), all references in our published articles must contribute to
the scholarly content of the paper and avoid bias (self-citations, journal citations, school of
thought, etc.,) and reflect the current state of knowledge in the field. We encourage you to
consider this and reduce the self-citations to make sure that only the most relevant citations
are kept.

We have reduced two self-citations and included additional references that are all red
highlighted in the revised manuscript.
3

Round 2

Reviewer 1 Report

Comments and Suggestions for Authors

In my opinion, the manuscript is suitable for publication in the present form.

Reviewer 2 Report

Comments and Suggestions for Authors

The authors have addressed satisfactorily my comments and the paper can now be published in its present form.